# The Association between the Mental Health Nurse-to-Registered Nurse Ratio and Patient Outcomes in Psychiatric Inpatient Wards: A Systematic Review

**DOI:** 10.3390/ijerph17186890

**Published:** 2020-09-21

**Authors:** Nompilo Moyo, Martin Jones, Diana Kushemererwa, Sandesh Pantha, Sue Gilbert, Lorena Romero, Richard Gray

**Affiliations:** 1School of Nursing and Midwifery, La Trobe University, Bundoora, VIC 3086, Australia; d.kushemererwa@latrobe.edu.au (D.K.); s.pantha@latrobe.edu.au (S.P.); s.gilbert@latrobe.edu.au (S.G.); R.Gray@latrobe.edu.au (R.G.); 2Department of Rural Health, University of South Australia, North Terrace, Adelaide, SA 5000, Australia; Martin.Jones@unisa.edu.au; 3Alfred Hospital, 55 Commercial Road, Melbourne, VIC 300, Australia; l.romero@alfred.org.au

**Keywords:** CRD42019145156, mental health, nursing, inpatient, skill mix, admission, relapse, empty review

## Abstract

Nursing skill mix in inpatient mental health wards varies considerably between countries. Some countries have an all-registered mental health nurse workforce; others have a mix of registered mental health and registered nurses. Understanding the optimal nursing skill mix in mental health inpatient units would inform service planning. This report aims to examine the association between the registered mental health nurse-to-registered nurse ratio and psychiatric readmission (or referral to community crisis services) in adult mental health inpatients. A systematic review was performed. We searched key databases for observational and experimental studies. Two researchers completed title-and-abstract and full-text screening. Our search identified 7956 citations. A full-text review of four papers was undertaken. No studies met our inclusion criteria. We report an empty review. Despite the obvious importance of the research question for the safe staffing of inpatient mental health services, there are no studies that have tested this association.

## 1. Introduction

### 1.1. Nursing Skill Mix in Medical and Surgical Wards

To date, there have been at least 21 observational studies that have tested the association between the educational preparation of nurses and patient outcomes [1,2,3,4,5,6,7,8,9,10,11,12,13,14,15,16,17,18,19,20,21]. Half of these studies have focused on surgical patients only. The remainder have been on medical, intensive care and specialist units. As far as we can determine, the first study to test the association between nursing skill mix and patient outcomes was reported in 1976 [22]. Most of these studies (15/20) are from an overlapping group of authors and have essentially followed the same methodology, i.e., estimating the skill mix, education and nurse-to-patient ratio by surveying nurses [11,16,21] and extracting aggregated patient mortality data from hospital administrative sources. Studies have generally reported a significant association between a more highly educated nursing workforce and a lower risk of mortality [12,15,16].

There are important methodological limitations that need to be considered when appraising these skill-mix studies, particularly around how skill mix is measured. Five studies [9,10,16,20,21] have reported using the same questionnaire to measure skill mix. While some authors have stated that the questionnaire has been validated, this is difficult to track in the published literature. We have been unable to get hold of a copy of the questionnaire from the original authors because of apparent copyright issues. It is likely that a self-reported measure of skill mix may lack precision, e.g., nurses asked about their qualifications may inflate their achievements.

Evidence from medical and surgical skill-mix studies cannot be generalised to mental health settings because nurses are working with different populations—typically, people with severe mental disorders—and the care required is different: a combination of psychosocial interventions, risk assessment, containment and medication management [23,24,25]. Psychosocial interventions are important in mental health and have been shown to be effective in reducing symptoms of anxiety, depression and posttraumatic stress disorder [26].

### 1.2. Nursing Skill Mix and Patient Outcomes in Mental Health Settings

One previous systematic review examined the association between mental health-nursing skill mix and relapse [27]. The review, focused on community mental health nurses and included two studies, involving 356 patients. The studies were rated as having a low–moderate risk of bias [27]. The authors reported that there was not enough evidence to conclude that community mental health nursing reduced the risk of admission to psychiatric inpatient facilities.

In the [27] review, admission and not mortality was used as the primary outcome of interest; this is because death is a comparatively rare outcome in mental health settings. Demonstrating an association between skill mix and mortality would require an unfeasibly large sample size. Aside from admission to psychiatric hospital, the other candidate outcomes that could be used in mental health-nursing skill-mix research are untoward incidents that include violence and aggression, absconding, seclusion and physical and mechanical restraint [28,29,30].

Untoward incidents have previously been used in evaluations of mental health service interventions. For example, Chou et al. [31] conducted an observational study to test if there was an association between patient, environmental and nurse staffing factors and assaults by patients on clinicians and patients in acute-inpatient psychiatric units. The authors included 287 patients and 855 assaults. The study showed an association between clinical experience and violence training and the risk of assault. However, the authors may have overestimated the actual number of assaults because they counted violence toward objects and verbal threats as assaults. This lack of consistency in how untoward incidents are defined and measured is a major limitation in their use as a valid measure in mental health research.

The use of untoward incidents (violence, aggression, restraint and seclusion) as an outcome in research requires careful consideration. Only a minority of people experiencing mental illness who are admitted to hospital will be violent or aggressive. If this is used as a research outcome, any observed association will likely only be in a small subgroup of patients (i.e., those that have been violent). There is also evidence that mental health clinicians working in different environments will have different views about what constitutes an untoward incident [32]. For example, the threshold for reporting violent incidents is considered higher in Psychiatry Intensive Care. Units (PICU), compared with in general acute-inpatient wards [32]. The number of untoward incidents may also change when practice on a ward is being observed as part of a research study or clinical audit. For example, if nurses are aware that there is a study ongoing on their ward where the outcome (a measure of success) is the number of untoward incidents, they may be less likely to report incidents because they want to please the researcher (a sort of social desirability bias). As a consequence, the number of reported incidents may seem to decline when there has, in fact, been no change [30].

### 1.3. Why Is Admission to Hospital a Good Research Outcome?

People who experience mental illness are admitted to inpatient care for the emergency treatment of acute psychiatric symptoms (hallucinations, delusions and mania) that are causing the patient distress and may be placing themselves or others at risk [33]. It has been argued that some admissions to psychiatric hospital are planned—e.g., for the initiation of clozapine. However, there is little evidence to demonstrate that this is a common occurrence [34,35]. Mental disorders are long-term conditions [36,37], and most people admitted to a psychiatric ward will have subsequent admissions [38,39,40]. For example, Schennach et al. [38] examined relapse and its predictors among patients with schizophrenia within one year of hospital discharge. Two hundred patients were included in the study. The authors reported that 104 patients (52%) had at least one relapse in the first year after hospital discharge, and 34 (17%) had multiple relapses. Among patients who relapsed, 78 (75%) were admitted as a result of the exacerbation of their symptoms [38].

Data on admissions may be reliable, as they are recorded as a part of routine hospital administrative work [41,42]. Admission data have been used by researchers internationally for many years in both medical and psychiatric settings [40,43]. However, there are notable limitations to using readmission as an outcome in mental health research. Patients may relocate or change residential address to a different catchment area and be admitted to a different hospital. While not a perfect measure, admission to a psychiatric hospital is a good proxy for relapse and is meaningful to patients, their family and friends, mental health services and the wider community [44,45,46].

### 1.4. Nursing Skill Mix on Mental Health Wards

A mental health nurse is a nurse who has completed specialist post-qualification education in mental health nursing, which builds upon primary generalist nursing education [47]. The nursing skill mix in mental health settings varies considerably from country to country. For example, in the UK, psychiatric wards are staffed exclusively by nurses that hold a specialist mental health registration [48,49]. Other countries—for example, Australia—have a mix of registered mental health and registered nurses working in a psychiatric inpatient unit [48,49]. Some countries have no mental health nurses working on psychiatric wards; nurses are registered but with no specialist education.

There is a need to investigate the association between nursing skill mix and patient outcome in inpatient psychiatric settings.

### 1.5. Aims

This systematic review aims to synthesize evidence examining the association between the mental health-to-registered nurse ratio and patient outcomes (relapse determined by hospital admission) in inpatient mental health settings.

### 1.6. Review Question

Is there an association between the registered mental health-to-registered nurse ratio and relapse in adult mental health inpatients?

## 2. Materials and Methods

We report Part Two of the registered report. Part One was published in the *Journal of Psychiatric and Mental Health Nursing* [50]. We therefore only present a summary of our methods. No changes were made to our study methodology subsequent to publishing Part One of this registered report. Our reporting adheres to the Preferred Reporting Items for Systematic Reviews and Meta-Analyses 2015 checklist (PRISMA) [51].

### 2.1. Search Strategy

We searched five key databases (MEDLINE, CINAHL, EMBASE, Cochrane Central and PsycINFO).

We used synonyms and related terms for the three key study concepts: mental health inpatients, nursing skill mix, and psychiatric nursing. Our search strategy shown in Table 1 was externally peer-reviewed. At least two reviewers (N.M., D.K. and S.P.) completed title-and-abstract and full-text screening. A third reviewer (R.G.) resolved any conflicts. Data extraction was also undertaken by two reviewers (N.M. and R.G.).

### 2.2. Eligibility Criteria

Our inclusion criteria were:Observational and experimental studies.Conducted in an acute-inpatient psychiatric unit.The manuscript was written in English.Involving patients aged 18 years or over.Reported registered mental health-to-registered nurse ratio as the measure of skill mix.Readmission to psychiatric inpatient care or referral to a mental health crisis team was a reported outcome.

We did not search for the grey literature because such studies have often not been through a rigorous peer-review process and the search strategies cannot be described in a replicable way.

## 3. Results

Figure 1 shows the flow of papers through the review process. Our initial search identified 7956 citations, of which 1811 were duplicates and were removed. A total of 6141 citations were excluded during the title-and-abstract screening. The full texts of four studies were reviewed [52,53,54,55] but rejected because they did not have the right exposure (registered nurse to registered mental health nurse ratio). Two of the studies also had the wrong outcome [54,55].

Since no studies were included, no detailed data synthesis or assessment of risk of bias in individual studies was performed.

### Excluded Studies

Empty reviews should summarise studies that were excluded at full-text screening, because these studies may provide relevant observations [56]. Table 2 summarises the key findings from the four studies excluded at full-text screening [52,53,54,55]. One study was not relevant to the aim of the review [52]—it examined the effect of industrial action by nursing staff on the clinical characteristics of admitted patients. The three other studies were focused on aspects of nursing skill mix and clinical outcomes in a mental health setting [53,54,55].

Han et al. [53] examined the association between nurse staffing and mental health care readmission within 30 days of discharge in 114 hospitals in South Korea. Data were extracted from the National Health Insurance claim dataset. The authors report an association between readmission rates (within 30 days of discharge) and the number of nurses working in the hospital. The odds of readmission were 5% lower for every 10 extra nurses per hospital. The [53] study provides some evidence that nurse skill mix may be associated with patient outcomes in inpatient mental health settings. However, the authors report few details about the accuracy and completeness of the data they extracted. For example, it is not clear how many missing data there were and whether there were systematic reasons why these data were missing. Important information about nurse staffing (e.g., age, qualifications, education and years of experience) were seemingly not extracted. These variables are likely confounders; the authors do not provide a commentary about why these key variables were omitted from the analysis. The authors’ failure to adequately adjust for confounding in this study severely restricts the value of the reported association.

De Lacy et al. [54] determined the association between nursing skill mix (nurse-to-patient ratio and ratio of registered and non-registered nurses) and rates of seclusion and restraint in six public psychiatric hospitals in the United States. The nurse-to-patient ratio for each participating hospital was calculated daily over the four years of the study. As far as we can determine, the nurse-to-patient ratio was calculated by dividing the number of full-time equivalent (FTE) nurses (registered and unregistered) by the average number of patients in the hospital over the study period. The use of the average number of inpatients as the denominator in this calculation may potentially lead to an over- or under-estimation of the actual nurse-to-patient ratio; this is because the number of inpatients may vary over time. For example, at weekends or during public holidays, there may be a reduction in the number of patients in the hospital. We wanted to check if our interpretation about how skill mix was calculated was correct, but contact details for the author were not reported in the publication (a university thesis). The registered-to-unregistered nurse ratio was calculated, again each day, by dividing the numbers of FTE-registered nurses by the total numbers of nurses working in the participating hospitals [54]. As with the nurse-to-patient ratio, the same caveats apply to how the skill mix was calculated.

Seclusion and restraint were the outcome in the De Lacy et al. [54] study. This was calculated at the hospital level by dividing the number of recorded seclusion events, again, by the average daily census number of patients. Still, the use of the likely average number of hospital patients impacts the precision of the measure. De Lacy et al. [54] do not report a significant association between the two skill-mix measures (the nurse-to-patient and registered-to-unregistered nurse ratios) and the use of seclusion and restraint. The authors state that there was a statistically significant association between both skill-mix measures and the number of episodes of seclusion and restraint. The authors acknowledged that important confounders were not adjusted for in their analysis (e.g., diagnosis, psychoactive medications, the numbers of scheduled individual and group treatments offered). As a consequence, the reported association needs to be interpreted with considerable caution.

Bowers et al. [55] conducted an observational study to determine whether an increase in nurse-staffing levels preceded or followed changes in the rates of conflict and containment in inpatient psychiatric wards. The study involved 32 acute wards in England. The authors used the Patient-Staff Conflict Checklist (PCC-SR) to measure rates of conflict and containment. The PCC-SR is an end-of-shift report completed by nurses about the frequency of conflict and containment (e.g., aggression, manual restraint, absconding and forced medication administration) events [56]. There are several potential sources of bias in the study [55] that need to be considered when interpreting the results. In the paper, the authors refer to the validity of PCC-SR, citing a paper by themselves [57]. In this paper, the author reported a statistically significant—but weak—correlation (*r* = 0.24) between the PCC-SR and “officially reported incidents” [57]. Whether the reported correlational data provide evidence that the PCC-SR is valid is a matter of conjecture. As we have already noted, it is generally accepted that “official reports” are likely to underestimate the actual number of incidents, i.e., the PCC-SR is a valid measure of a measure that is generally considered not to be valid [58,59,60]. In the Bowers et al. [55] study, the authors selected a sub-sample of 32 of the 136 participating wards for this secondary analysis. The selected wards had a high level of compliance with completing the PCC-SR at the end of the shift. It is not stated—though it seems likely—that this analysis was post hoc and not planned a priori. The authors confirm that the included wards were different to the sample as a whole; for example, they had higher bed numbers. Bowers et al. [55] downplay the differences between the two groups by using temperate language (e.g., slightly higher and marginally lower) in the results section of their manuscript. The key observation in the [55] study was that the more nurses working on a shift, the greater the number of incidents. The most likely explanation for the reported association is bias; consequently, we argue, the observations should be treated with some caution.

## 4. Discussion

This systematic review aimed to identify and synthesize evidence for an association between the mental health-to-registered nurse ratio and relapse in inpatient mental health settings. No studies met our inclusion criteria; consequently, we report an empty review. We have identified the need for high-quality skill-mix research to determine optimal and safe staffing in mental-inpatient settings.

An empty review is a systematic review that finds no primary studies eligible for inclusion [61]. Empty reviews are common; for example, in 2010, 9% of the 4320 reviews published in the Cochrane Library were empty reviews [61]. Paper [62] reported that there is some evidence to suggest that authors do not submit empty reviews because they perceive they will be challenging to get published. Empty reviews make an essential contribution to our understanding of problems because they identify gaps in knowledge that need to be addressed in future research [61].

While there have been many extensive studies that have established an association between nursing skill mix and patient outcomes [28,31,54], few studies have examined the effect of mental health nursing skill mix and patient outcomes. There is not a complete absence of nursing skill-mix research in mental health. For example, we noted two studies that have tested the associations between the nurse-to-patient ratio and patient outcomes—typically, the number of incidents of violence and aggression—in inpatient mental health settings (e.g., [53,55]). Readmission has been used in both general wards and psychiatric wards as an outcome in skill-mix studies. For example, [53] reported that psychiatric hospitals with higher nurse-staffing numbers were associated with lower rates of admission.

### Review Limitations

There are some limitations to our review that are important to consider. It could be argued that we should have undertaken a scoping review of the mental health nursing skill-mix research to inform the viability of a systematic review. A scoping review is a type of review that provides a preliminary assessment of the potential size and scope of the available research literature [63]. It can inform researchers as to whether a full systematic review is needed. Scoping reviews cannot be regarded as a final output in their own right, mainly because limitations in their rigour and their duration mean that they hold the potential for bias [63].

In this review, our outcome of interest was admission to acute services. This is an outcome that is extensively used in mental health services research [46,64]. Skill-mix research in medical and surgical settings has tended to use mortality as an outcome [3,65]. This would not be appropriate in mental health settings because death is a rare—albeit not unheard of—event. Untoward incidents (e.g., the number of violent incidents) is an alternative measure but is essentially impossible to measure in a valid way. The definition and reporting of untoward incidents vary among hospitals, making it impossible to aggregate them as a meaningful measure. For example, a hospital can report an incident as an untoward incident, but the same may not be reported as an untoward incident by another hospital. The untoward incidents in hospitals are underreported [58,59,60]. We did identify studies where the number of untoward incidents was the reported outcome. There may be merit in undertaking a systematic review where untoward incidents are the outcome of interest.

In this review, we focused on a single primary outcome. We could have included skill-mix studies with any outcome. However, there is a considerable risk of selective-reporting bias, where the study authors select, post hoc, the most interesting outcomes [66].

Our review was restricted to the English language and did not include grey literature. If there are unpublished studies or studies from other languages regarding our topic, they might have been missed.

## 5. Conclusions

There are no peer-reviewed studies that test the association between the registered mental health-to-registered nurse ratio and psychiatric readmission (or referral to a mental health crisis service) in adult psychiatric inpatients. This is a gap in knowledge that would benefit from further investigation.

## Figures and Tables

**Figure 1 ijerph-17-06890-f001:**
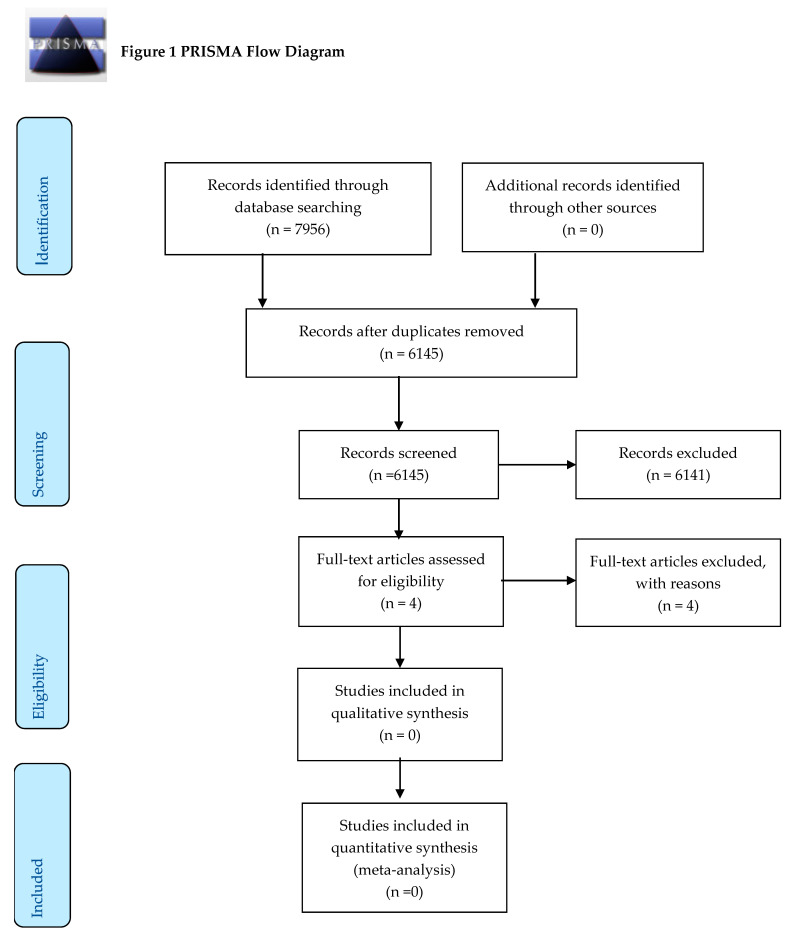
Preferred Reporting Items for Systematic Reviews and Meta-Analyses 2015 (PRISMA) flow diagram.

**Table 1 ijerph-17-06890-t001:** Database(s): Ovid MEDLINE(R), ALL 1946 to 30 July 2019.

Search Strategy:	
**#**	**Searches**
1	exp Hospitals, Psychiatric/
2	exp Mental Disorders/
3	exp Psychiatric Department, Hospital/
4	exp Mental Health/
5	(acute mental unit * or psychiatric inpatient care or psychiatric ward * or mental ward * or mental health unit *).mp.
6	(psychiatric service * or mental health care or psychiatric care or mental health care).mp.
7	(mental health hospital or psychiatric institution * or mental health institution * or mental health setting * or psychiatric setting * or psychiatric hospital *).mp.
8	(mental health inpatient or psych * inpatient * or psych * patient * or mentally ill patient * or mental patient or mental health patient *).mp.
9	1 or 2 or 3 or 4 or 5 or 6 or 7 or 8
10	(nurse patient ratio * or nursing staffing ratio * or nursing skill ratio * or nursing skill mix * or hours per patient day or nurse per patient day or full time equivalent or nurs * staff * or nurs * schedul * or task allocation * or delegation).mp.
11	(nursing staff numbers or staff mix or staffing levels).mp.
12	(educational preparation or education level or nursing care or grade mix or nurs * staff mix * or nursing grade * or care hours per patient *).mp.
13	exp Nursing Staff, Hospital/or Nursing/
14	11 or 12 or 13
15	exp Psychiatric Nursing/
16	(mental health registered nurs * or mental health nurs * or psych * nurs *).mp.
17	15 or 16
18	9 and 14 and 17
19	limit 18 to English language

* truncation.

**Table 2 ijerph-17-06890-t002:** Summary of the studies excluded at full-text screening.

Author	Setting	Design	Exposure	Measure of Exposure	Outcome	Results	Reason for Exclusion
Han et al., 2015 [53]	South Korea	Observational study	Psychiatrists and nurse staffing	Number of nurses Proportion of experienced psychiatrists	Readmission within 30 days of discharge	The odds of readmission were 5% lower for every 10 extra nurses per hospital.	Wrong exposure
Abdelkader et al., 1990 [52]	England	Comparison study	Nurses’ industrial action	Six-month period during the industrial action in 1982	Admission	In 1982, there was a reduction in the total number of admissions by 30% compared to 1981.	Wrong exposure
Bowers et al., 2012 [55]	England	Time series analysis	Nursing staff numbers	Numbers of nursing staff on duty	Conflict and containment incidents	An increase in qualified nurses working on a ward was associated with a subsequent increase in conflict and containment incidents.	Wrong exposure and outcome
De Lacy et al., 2006 [54]	United States	Descriptive correlational research	Nursing staff numbers and skill mix	Nursing care staffing Nursing staff numbers Nursing skill mix Full-time equivalents	Seclusion and restraint use	An increased proportion of registered nurses was associated with a decrease in seclusion- and restraint-use measures.	Wrong exposure and outcome

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
