# Peer review of "The Association between the Mental Health Nurse-to-Registered Nurse Ratio and Patient Outcomes in Psychiatric Inpatient Wards: A Systematic Review"

_ijerph, 2020, doi:10.3390/ijerph17186890_

Round 1
Reviewer 1 Report
Dear Authors:
Thanks by your work in this area. It`s a very interesting.
Some comments:
- You mentioned that different countries have a different requirements about the nurse mix skill. Did you know if there is data about different outcomes for patients between those countries? I think that this kind of information can give to the readers a very interesting point that support the investigation.
- Seems obvious to me that the nursing approach with patients in a med-surg ward is so different that approach in a psychiatric ward. In some way, the nurses needs to change their approach because the nature of the care it´s different. A nurse that was trained in generals skills, in some years of work in a psychiatric ward, surely made adjusts. So, in your arguments, what is your theory about the different process and care that mental health nurse can offer to patients comparing with a nurse with a general training?
- Your reasons about the outcome that select are very convincing.
- I seems to me that is no good write that you sent this work to another journal and was rejected by the editor without explanation. I think that some readers could think that is a complaint.
Discussion about to published an empty review is a very good discussion, and I think that the reasons to make it is related with the importance of the theme.
I agree with you that expose that is no evidence yet, is a first step.
Best regards,
Author Response
Response to Reviewer 1 Comments
Point 1: You mentioned that different countries have a different requirements about the nurse mix skill. Did you know if there is data about different outcomes for patients between those countries? I think that this kind of information can give to the readers a very interesting point that support the investigation.
Response 1: We could not find data on the effect of mental health nursing skill mix and education on psychiatric inpatient outcomes.
Point 2: Seems obvious to me that the nursing approach with patients in a med-surg ward is so different that approach in a psychiatric ward. In some way, the nurses needs to change their approach because the nature of the care it´s different. A nurse that was trained in generals skills, in some years of work in a psychiatric ward, surely made adjusts. So, in your arguments, what is your theory about the different process and care that mental health nurse can offer to patients comparing with a nurse with a general training?
Response 2: General nurse training does not include psychotherapies, comprehensive mental state examination and psychopharmacology.
Point 3: Your reasons about the outcome that select are very convincing.
Response 3: Thank you.
Point 4: I seem to me that is no good write that you sent this work to another journal and was rejected by the editor without explanation. I think that some readers could think that is a complaint.
Response 4: This sentence was deleted “Part two was submitted on 11 June 2020 and was rejected by the editor without explanation”
The new statement reads: “We are reporting part two of the registered report. Part one was published in the journal of psychiatric and mental health nursing [49]. We therefore present a summary of our methods. No changes were made to our study methodology subsequent to publishing part one of this registered report. Our reporting adheres to the Preferred Reporting Items for Systematic Reviews and Meta-Analyses checklist 2015 (PRISMA) [50].”

Reviewer 2 Report
Although the authors declare that no studies met the inclusion criteria and consequently, the final report is an empty review, the result is still significant.
The systematic review aimed at identifying evidence testing the association
between the mental health to registered nurse ratio and relapse in inpatient mental health settings, and the final considerations highlight the need for studying this aspect, missing in the literature.
English language and typos are to be well checked.
Author Response
Reviewer 2
Point 1: Although the authors declare that no studies met the inclusion criteria and consequently, the final report is an empty review, the result is still significant.
Response 1: Thank you.
Reviewer 3 Report
The authors conducted a systematic review to evaluate the association between the mental health nurse to registered nurse ratio and clinical outcome in patients with mental disorders, based on the hypothesis that nurses who has completed specialist post-qualification education in mental health nursing might positively impact patient outcome.
Despite being of high relevance, this topic has not been the object of previous studies, leading the authors not to be able to identify any study meeting the inclusion criteria.
The manuscript is clear and well written. The rationale is well explained and an important strength of this review include the presence of a registered protocol.
Despite the important limitation of being an empty review, the discussion of results and limitations of excluded articles still adds to the scarce literature on this topic and provides a relevant contribution.
I have a few minor suggestions:
- the authors should report in the present manuscript a complete search strategy for at least one database
- please add initials of authors who screened the reviewed articles, as well as the third author that risolved conflicts
- can the authors hypothesize why, among such as a high number of retrieved citations, only four articles were potentially relevant (i.e. were not excluded based on title and abstract)?
Reviewer 4 Report
I consider that it is not necessary to indicate in the manuscript that it was rejected in another journal, with indicating the reference for the reading of the methodology is sufficient.
Describing the skills of mental health nurses could help to understand why they have an impact on the outcome
The authors say that they could have included skill mix studies with any outcome, 4 excluded studies with different outcome are described, why choose these and not others with different outcome?
